# Explainable AI-based Alzheimer's prediction and management using multimodal data

**Sobhana Jahan**[1,2], **Kazi Abu Taher**[2], **M. Shamim Kaiser**[3]*, **Mufti Mahmud**[4], **Md. Sazzadur Rahman**[3], **A. S. M. Sanwar Hosen**[5], **In-Ho Ra**[6]*

1 Department of Computer Science and Engineering, Bangladesh University of Professionals, Dhaka, Bangladesh, 2 Department of Information and Communication Technology, Bangladesh University of Professionals, Dhaka, Bangladesh, 3 Institute of Information Technology, Jahangirnagar University, Savar, Dhaka, Bangladesh, 4 Department of Computer Science, Nottingham Trent University, Nottingham, United Kingdom, 5 Department of Artificial Intelligence and Big Data, Woosong University, Daejeon, South Korea, 6 School of Computer, Information and Communication Engineering, Kunsan National University, Gunsan, South Korea

* mskaiser@juniv.edu (MSK); ihra@kunsan.ac.kr (IHR)

**Data Availability Statement:** Open source Data available at https://oasis-brains.org/.

**Funding:** This work is supported by the research funds of the ICT division of the Government of the

## Abstract

### Background

According to the World Health Organization (WHO), dementia is the seventh leading reason of death among all illnesses and one of the leading causes of disability among the world's elderly people. Day by day the number of Alzheimer's patients is rising. Considering the increasing rate and the dangers, Alzheimer's disease should be diagnosed carefully. Machine learning is a potential technique for Alzheimer's diagnosis but general users do not trust machine learning models due to the black-box nature. Even, some of those models do not provide the best performance because of using only neuroimaging data.

### Objective

To solve these issues, this paper proposes a novel explainable Alzheimer's disease prediction model using a multimodal dataset. This approach performs a data-level fusion using clinical data, MRI segmentation data, and psychological data. However, currently, there is very little understanding of multimodal five-class classification of Alzheimer's disease.

### Method

For predicting five class classifications, 9 most popular Machine Learning models are used. These models are Random Forest (RF), Logistic Regression (LR), Decision Tree (DT), Multi-Layer Perceptron (MLP), K-Nearest Neighbor (KNN), Gradient Boosting (GB), Adaptive Boosting (AdaB), Support Vector Machine (SVM), and Naive Bayes (NB). Among these models RF has scored the highest value. Besides for explainability, SHapley Additive exPlanation (SHAP) is used in this research work.

People's Republic of Bangladesh in 2020 – 2021 (Fiscal year), the Woosong University Academic Research Fund in 2023. The founders had no role in study design and data collection and analysis. We have no conflict of interest to disclose.

**Competing interests:** The authors have declared that no competing interests exist.

## Results and conclusions

The performance evaluation demonstrates that the RF classifier has a 10-fold cross-validation accuracy of 98.81% for predicting Alzheimer's disease, cognitively normal, non-Alzheimer's dementia, uncertain dementia, and others. In addition, the study utilized Explainable Artificial Intelligence based on the SHAP model and analyzed the causes of prediction. To the best of our knowledge, we are the first to present this multimodal (Clinical, Psychological, and MRI segmentation data) five-class classification of Alzheimer's disease using Open Access Series of Imaging Studies (OASIS-3) dataset. Besides, a novel Alzheimer's patient management architecture is also proposed in this work.

## Introduction

Alzheimer's Disease (AD) is a chronic, progressive neurodegenerative disease that gradually deteriorates memory and cognitive abilities, and it is the most common cause of dementia in older people. In the world, there were more than 50 million AD sufferers in 2018 [1]. Worldwide, there will be 131 million persons with AD in 2050, and the socioeconomic cost will be 9.12 trillion dollars [2]. Forgetting recently acquired information, important dates or events, difficulty in performing simple daily works, and repeatedly asking the same questions are all classic early symptoms of AD. In the final stage, patients' behavioral changes are also observed. The disease strikes the majority of people in their mid-60s. Scientists agree that the root cause of this neurological disease is a combination of genetics, long-term environmental conditions, and lifestyle [3]. Though some medications are available, AD is not curable, and the damage it causes is permanent. The most common cause of death in Alzheimer's patients is aspiration pneumonia [4].

Though machine learning (ML) is a very potent technique in AD diagnosis, patients nowadays are more likely to visit clinics and have their AD diagnosed. Even doctors are willing to rely on clinical diagnosis rather than the ML model's prediction. A study shows participants (176 persons) were less likely to trust Artificial Intelligence (AI) or ML models for diagnosis and treatment than doctors. Even when participants are informed that ML outperforms a human doctor, their trust in ML does not grow [5]. The main reason behind this distrust is the black-box nature of ML models. For these reasons, explaining the ML models' decision or the features responsible for this decision is the best way to gain the trust of the people. Explainable AI is a tool that can annotate a model's decisions as well as decision-making characteristics. Furthermore, the use of only neuroimaging data for AD prediction is very common, and achieving good accuracy is difficult in most cases. Even using single modal data for this type of critical prediction is very risky as sometimes it can produce faulty predictions. To remove these problems, our work proposes an explainable ML model using multimodal data.

The quality of life of AD patients can be improved if there is a constant helping hand with them. But, due to modernization and capitalism, almost all family members are busy with their work. Due to the lack of care and support, the condition of those patients may decrease very rapidly. Besides, the scarcity of Alzheimer's patient care centers is also a problem. A sensor-based and IoT-enabled real-time monitoring and AD patient management system can provide a cost-effective way of continuous monitoring, support for daily activity, early warning of health deterioration, and emergency medical care [6, 7]. For this reason, this paper presents an Alzheimer's patient monitoring and management framework.

To the best of our knowledge, we are the first to present this explainable multimodal approach using clinical data, neuroimaging data, and psychological data, which are collected from the Open Access Series of Imaging Studies (OASIS-3) dataset [8]. The main contributions of this research work are:

a. A five-class AD prediction approach using a multimodal dataset is proposed. To ensure multimodality, data-level fusion using Alzheimer's Disease Research Center (ADRC) clinical data, brain Magnetic Resonance Imaging (MRI) segmentation data, and psychological assessments are performed.

b. The black box ML decisions are converted to an explainable one where general people can understand the reasons behind any prediction.

c. A 24/7 AD patient monitoring and management system framework is also proposed.

The rest of this paper is organized as follows: some recently published similar works, the methodology of the proposed work, the performance analysis, and the concluding remarks.

## Related works

Our research work completely depends on multimodal AD prediction using explainable AI. There are some relevant published research works in this field using MRI, PET, gene data, medical history, neuropsychological battery, and cognitive scores data. For the explanation of the model, Local Interpretable Model-agnostic Explanations (LIME) and Shapley Additive explanations (SHAP) are powerful tools. Some relevant research works using above mentioned dataset and tools are described below.

Multidirectional Perception-Generative Adversarial Networks (MP-GAN), a multidirectional mapping method that Yu et al. [9] incorporated is capable of effectively capturing the salient global features. Thus, the proposed model can clearly differentiate the subtle lesions via MR image transformations between the source domain and the target domain by using the class discriminative map from the generator. A single MP-GAN generator may also learn the class-discriminative maps for several classes by combining the classification loss, cycle consistency loss, adversarial loss, and L1 penalty. According to experimental findings on the available Alzheimer's Disease Neuroimaging Initiative (ADNI) dataset, MP-GAN can precisely depict numerous lesions that are impacted by the development of AD.

Lei et al. [10] created a deep learning and joint architecture to forecast clinical AD scores. Particularly, to minimize dimensions and screen the features of brain regions associated with AD, the feature selection method integrating group LASSO and correntropy is applied. To investigate the internal connectivity between various brain regions and the temporal correlation between longitudinal information, we investigate the multi-layer independently recurrent neural network regression. The suggested joint deep learning network analyzes and forecasts the clinical score by looking at the correlation between MRI and clinical score. Doctors can perform an early diagnosis and prompt treatment of patient's medical conditions using the projected clinical score values.

To evaluate Mild Cognitive Impairment (MCI) and AD, Yu et al. [11] suggested a unique tensorizing GAN using high-order pooling. The suggested model may make maximal use of the second-order statistics of integrative MRI by including the high-order pooling technique in the classifier. The first attempt to cope with categorization on MRI for AD diagnosis is the suggested Tensor-train, High-order Pooling, and Semi-Supervised GAN (THS-GAN). Extensive experimental findings on the ADNI data set show that the THS-GAN performs better

when compared to current approaches. It also demonstrated how pooling and tensor training might enhance the classification results.

Using MRI images and gene data sets, Kamal et al. [12] used SpinalNet and Convolutional Neural Network (CNN) to classify AD from MRI images. The researchers then used microarray gene expression data to classify diseases using K-Nearest Neighbor (KNN), Support Vector Classification (SVC), and XG boost classification techniques. Instead of using gene and image data solely, the authors had combined these two approaches and explained the results using the LIME method. For MRI image classification, the accuracy rate of CNN is 97.6%, and for the gene expensive data, SVC outperforms other approaches.

El-Sappagh et al. [13] proposed a two-layered explainable ML model for AD classification. It was a multimodal approach where data from 11 modalities (genetics, medical history, MRI, Positron Emission Tomography (PET), neuropsychological battery, cognitive scores, etc.) were integrated. Random Forest (RF) classifier was used here in the first layer for multiclass classification, and the results were explained using the SHAP framework. In the second layer, binary classification took place, where probable MCI to AD classification took place. For the first layer, the achieved cross-validation F1-score was 93.94%, and accuracy was 93.95% (multi-class classifier). For the second layer, accuracy and F1-score were 87.08% and 87.09%, respectively. The accuracy and the F1 score of the model could be increased if the authors try to use the deep learning technique. Besides, this model obtained satisfactory performance from an academic point of view not from real-life.

Lee et al. [14] proposed a multimodal Recurrent Neural Network (RNN) model to predict AD from the MCI stage. In this approach, the authors had integrated subjects' longitudinal Cerebrospinal Spinal Fluid (CSF) and the cognitional performance biomarkers along with cross-sectional neuroimaging data and demographic data. Here, all data were collected from the ADNI website. The proposed model was divided into two layers. Layer one consists of four Gated Recurrent Units (GRUs) where each contains one modality of data. From the 1st layer, a fixed-sized feature vector was produced. Then, the vectors were concatenated to the input for the final layer. The final layer presents the ultimate prediction. From MCI to AD prediction, the proposed model achieved 76% accuracy and 0.83 AUC using data from a single modality, whereas 81% accuracy and 0.86 AUC value had been achieved using multimodal data. The parameter optimization for the second training stage did not affect the parameters in every GRU for feature extraction, so each GRU could not learn from the final classification based on the collective features, which is a limitation of this model. A possible solution to this problem is to link GRUs to logistic regressions with in second step. Furthermore, the structure of this model should be designed in such a way that individual GRU components can extract integrative features.

Zhang et al. [15] proposed a multimodal multi-task learning method for predicting multiple features from multimodal data. This method was divided into two parts. First, a multi-task feature selection that selects a common subset of relevant features for multiple variables from each modality. Second, a multi-modal support vector machine that fuses the previously selected features from all modalities to predict multiple variables. Here, all data were collected from the ADNI website. The accuracy of the proposed model was 83.2%±1.5% (MCI vs HC) and 93.3%±2.2% (AD vs HC). One drawback of this method was that it was based on multimodal data, such as MRI, PET, and CSF, and thus necessitates each subject to had the corresponding modality data, limiting the number of participants that could be studied. For example, the ADNI database contains approximately 800 participants, but only about 200 of them have all baseline MRI, PET, and CSF data.

Baglat et al. [16] used ML techniques such as Logistic Regression (LR), Decision Tree (DT), Random Forest (RF), Support Vector Machine (SVM), and AdaBoost to improve the early

diagnosis and classification of Alzheimer's disease in the OASIS MRI dataset, with the RF classifier outperforming the others. The RF classifier's accuracy, recall, and area under coverage values are 86.8%, 80%, and 0.872, respectively. Ali et al. [17] employed an ensemble learning-based modified RF to detect AD. Using the OASIS-2 dataset, the suggested approach attained an accuracy of 96.43%. Kavitha et al. [18] proposed RF classifier to predict AD and achieved 86.92% accuracy using MRI OASIS dataset.

Amrutesh et al. [19] employed two separate datasets for the detection of AD: the longitudinal dataset, which contains text values, and the OASIS dataset, which contains MRI pictures. The OASIS Longitudinal dataset was used to train on 14 ML algorithms, including the RF Algorithm, which has a maximum accuracy of 92.1385% and a baseline accuracy of 47.1910%. Buvari et al. [20] proposed a CNN and Neural Network (NN) (dense) based AD prediction multimodal model where MRI and Numerical Freesurfer MRI segmentation data were used. Dataset was collected from the OASIS-3 website. The accuracy of numerical, image and hybrid approaches was 73.593%, 71.429%, and 74.891%. The performance of these research works was not excellent. Also, these models were black-box in nature.

From all these research works it is clearly understandable that the use of multimodal data can be a good way to towards better model's performance. SHAP would be a powerful tool to interpret the decision-making features of a model.

## Proposed alzheimer's prediction model

### Dataset acquisition and preparation

The Open Access Series of Imaging Studies (OASIS)-3 [8] aims to make neuroimaging datasets freely accessible to the scientific community. The OASIS-3 dataset includes longitudinal neuroimaging, cognitive, clinical, and biomarker data for normal aging and AD. This dataset contains the data of people from 42 to 95 years old.

Here, all participants data were provided with an identifier, and all dates were deleted and standardized to reflect the days since their enrollment. Many of the Magnetic Resonance (MR) sessions were accompanied by volumetric segmentation files generated by Freesurfer. For this research work, we have used the ADRC clinical data, psychological data, and Freesurfer volumetric MRI segmentation data. Table 1 shows the total number of data in each modality along with number of features.

**ADRC clinical data.** ADRC clinical data [8] consists of the longitudinal data of unique 1098 participants. The ADRC clinical data consists of the longitudinal data of unique 1098 participants. Number of CN, AD, Other dementia/Non AD, Uncertain, and Others are 4476, 1058, 142, 505, and 43, respectively.

There are various features of cognitively normal, AD dementia, uncertain dementia, and some non-AD dementia. For non-AD dementia label, Vascular Dementia, Dementia with Lewy Bodies Disease (DLBD), PD, and Frontotemporal dementia are considered. The important features of this dataset were Mini-Mental State Exam (MMSE), age, Judgment, memory, APOE (apolipoprotein E gene), Personal care, height, weight, Orient (recent and long-term

**Table 1. Description of multimodal dataset.**

| Data Modality | Unique Participants | Longitudinal Data | Number of Features |
|---|---|---|---|
| Clinical Data | 1098 | 6224 | 13 |
| MRI segmentation Data | 1053 | 2047 | 10 |
| Psychological Assessment data | 810 | 3342 | 17 |

memory testing), Clinical Dementia Rating (CDR), and Sumbox (clinical dementia rating scale). The ADRC administered a battery of neuropsychological tests to participants aged 65 and up every year. The Mini-Mental State Examination (MMSE) is a test that assesses general cognitive function, with scores ranging from 0 (severe impairment) to 30 (no impairment).

Clinical assessment protocols were completed by participants in accordance with the National Alzheimer Coordinating Center Uniform Data Set (UDS). UDS assessments included a medical history, a physical examination, and a neurological evaluation. Every three years, participants aged 64 or younger underwent clinical and cognitive assessments. Participants aged 65 and up had annual clinical and cognitive evaluations. The Clinical Dementia Rating (CDR) Scale was used to assess dementia status for the UDS, with CDR 0 indicating normal cognitive function, CDR 0.5 for very mild impairment, CDR 1 for mild impairment, and CDR 2 for moderate dementia; once a participant reached CDR 2, they were no longer eligible for in-person assessments. Age at entry, height, weight, and CDR evaluations are all included in the OASIS datatype "ADRC Clinical Data" (UDS form B1 and B4 variables). Clinicians completed a diagnostic impression intake and interview as part of the assessment, which resulted in a coded dementia diagnosis that was recorded in the OASIS datatype "ADRC Clinical Data". "Cognitively normal," "AD dementia," "vascular dementia," and contributing factors such as vitamin deficiency, alcoholism, and mood disorders are all diagnoses for this variable. The diagnostic determination for variables dx1-dx5 is distinct from UDS assessments, but diagnostic conclusions may overlap.

The Clinical Dementia Rating (CDR) is a global rating device that was developed in 1982 for a prospective study of patients with mild "senile dementia of AD type" (SDAT). Later, new and revised CDR scoring rules were introduced by Berg, 1988 and Morris, 1993. CDR is determined by testing six different cognitive and behavioral domains, including memory, Orientation, Judgment and problem solving, community affairs, home and hobby performance, and personal care. The CDR is a scale of 0–3 that includes no dementia (CDR = 0), questionable dementia (CDR = 0.5), MCI (CDR = 1), moderate cognitive impairment (CDR = 2), and severe cognitive impairment (CDR = 3). CDR itself is an outcome or target label for AD prediction. To make an unbiased dataset, we are removing this feature from the dataset.

**Psychological assessment data.** The psychological assessment dataset [8] contains various popular psychological tests such as Boston naming test, Trailmaking A (Trail A), Trailmaking B (Trail B), animals, vegetables, digit symbol, digit span, logical memory, Wechsler Adult Intelligence Scale (WAIS), and so on. Digit Span tests participants' attention and working memory by having them repeat a series of digits forward and backward. The number of trials correctly repeated forward and backward, as well as the longest length the participant can repeat back, were used to determine the participant's score. The Category Fluency Test, which requires participants to name as many words belonging to a category, such as animal and vegetable, and the Boston Naming Test, which requires participants to name drawings of common objects, were used to assess semantic memory and language. The WAIS-R Digit Symbol test and the Trail Making Test Part A were used to assess psychomotor speed. The number of digit symbol pairs completed in 90 seconds is used to score the WAIS-R Digit Symbol test. The Trail Making Test Part B was used to assess executive function. Participants in the Trail Making Test were asked to connect a series of numbers (1-26) for part A and a series of alternating numbers and letters (1-A-2-B) for part B to create a trail. Total time to complete in seconds, with a maximum of 150s for Trails A and 300s for Trails B, number of commission errors, and number of correct lines are all outcome measures. The Wechsler Memory Scale-Logical Revised's Memory—Story A measures episodic memory. Participants are asked to recall as many details as possible from a short story containing 25 bits of information after the examiner

reads it aloud and again after a 30-minute delay, with scores ranging from 0 (no recall) to 25 (complete recall).

**MRI segmentation data.** Freesurfer is an open-source software suite that can process and analyze MRI images of the human brain. This Freesurfer dataset [8] gives us the value of volumetric data of different parts of the human brain such as the intracranial, total cortex, left and right hemisphere cortex, subcortical gray, total gray, supratentorial, left and right hemisphere cortical white matter, and cortical white matter.

Using either FreeSurfer v5.0 or v5.1, T1-weighted pictures were processed. This processing includes motion correction and averaging of volumetric T1 weighted images, removal of non-brain tissue using a hybrid surface/watershed deformation procedure, automated Talairach transformation, segmentation of the subcortical white white matter and deep gray matter volumetric structures (including the hippocampus, caudate, putamen, amygdala, and ventricles) intensity normalization, tessellation of the gray matter white matter boundary, and automated topology correction. Following the completion of the cortical models, a number of deformable procedures were performed for further data processing and analysis, including surface inflation, registration to a spherical atlas based on individual cortical folding patterns to match cortical geometry across subjects, parcellation of the cerebral cortex into units with respect to gyral and sulcal structure, and creation of a variety of surface based data, including maps of curvature and sulcal depth. In segmentation and deformation procedures, this method uses both intensity and continuity information from the entire three-dimensional MR volume to produce representations of cortical thickness, calculated as the closest distance from the white/gray boundary to the CSF/gray boundary at each vertex on the tessellated surface.

In segmentation and deformation procedures, this method uses both intensity and continuity information from the entire three-dimensional MR volume to produce representations of cortical thickness, calculated as the closest distance from the white/gay boundary to the CSF/gray boundary at each vertex on the tessellated surface. Because the maps are generated using spatial intensity gradients across tissue classes, they are not solely dependent on absolute signal intensity. The maps generated are not limited to the voxel resolution of the original data and can detect submillimeter differences between groups. Cortical thickness measurement procedures have been validated against histological analysis and manual measurements. The test-retest reliability of Freesurfer morphometric procedures has been demonstrated across scanner manufacturers and field strengths.

## Multimodal dataset pre-processing

This research work consists of AD prediction using multimodal data. Here, data-level fusion is performed for creating multimodal data. For making it multimodal, we have integrated three different domain datasets which are the clinical data, the MRI segmentation data, or more precisely, brain MRI segmentation data, and the psychological assessment data. The data fusion process begins with the integration of three individual datasets. This integration has taken place in the OASIS website. This website provides the facility to join different domain data of each patient based on subject ID and session.

**The number of instances in each modality.** After creating the joined dataset, Psychological and Clinical assessments contain 3342 instances of the same participants. But the MRI segmentation dataset contains 3220 instances. So, in the MRI segmentation dataset the number of missing instances are (3342-3220) = 122. The number of unique participants: 810 (Psychological and Clinical) and 799 (MRI segmentation).

**Labels in multimodal dataset.** In the joined dataset, there are total 34 types of labels. All 34 types of labels can be categorized into 5 major types. Those 5 types are: CN, AD, non-AD,

**Table 2. Five major labels, their corresponding labels, and number of instances.**

| Major Labels | Labels | Instances |
|---|---|---|
| CN | CN | 2248 |
| AD | AD Dementia, AD dem w/depresss- not contribut, AD dem distrubed social- with, AD dem visuospatial- with, AD dem Language dysf after, AD dem w/PDI after AD dem not contrib, AD dem distrubed social- prior, AD dem w/PDI after AD dem contribut, AD dem w/oth (list B) contribut, AD dem w/depresss- contribut, AD dem Language dysf prior, AD dem Language dysf prior, AD dem w/oth (list B) not contrib, AD dem w/CVD contribut, AD dem cannot be primary, AD dem Language dysf with, AD dem w/CVD not contrib, AD dem w/oth unusual feat/subs demt, AD dem w/depress contribut, AD dem distrubed social- after, AD dem visuospatial- after, AD dem w/oth unusual features/demt, and AD dem w/depresss not contribut | 669 |
| Non AD | Non AD dem- Other primary, Vascular Demt- primary, Incipient Non-AD dem, Frontotemporal demt. prim, DLBD- primary, Incipient demt PTP, Dementia/PD- primary, DLBD- secondary, and Vascular Demt- secondary. | 106 |
| Uncertain Dementia | Unc: ques. Impairment, Unc: impair reversible, uncertain possible Non AD dementia, and uncertain dementia | 287 |
| Others | Dot, NULL, and 0.5 in memory | 26 |

uncertain dementia, and others. In Table 2, all 34 types of labels and their 5 major labels are clearly stated along with number of instances in each label.

## Missing data imputation using KNN

Missing data imputation using K-Nearest Neighbors is used here to fill up the missing values. The training set's N nearest neighbors are used to compute the mean value for each sample's missing values. Two samples are close if the features that neither is missing are close. Table 3 shows the root mean dange value (RMSE) of imputed dataset using different number of neighbors. For 2 neighbors the RMSE value is lower than others. So, the joined or fused dataset is imputed with 2 neighbors.

## Statistical analysis of multimodal dataset

The joined dataset or the multimodal dataset contains total 39 features. The value of mean, median, mode, and standard deviation of these 39 features are stated in Table 4. It is clear that the mean, median, and mode of these features are very close to each other except TrailA, TrailB, WAIS, and weight. As, these feature values are real medical data, small fluctuation from the expected result is acceptable. Even, Pearson correlation has verified that those features are necessary for creating the model. These are the reasons behind considering TrailA, TrailB, WAIS, and weight.

**Table 3. RMSE value after data imputation using KNN.**

| Neighbors Number | RMSE |
|---|---|
| **2 Neighbors** | **0.806** |
| 3 Neighbors | 0.834 |
| 5 Neighbors | 0.904 |

**Table 4. Statistical analysis of features.**

| Feature | Mean | Median | Mode | Standard Deviation |
|---|---|---|---|---|
| Logimem | 12.25059844 | 13 | 13 | 5.223761346 |
| Digif | 8.278874925 | 8 | 8 | 2.094828747 |
| DigiFlen | 6.599640934 | 7 | 6 | 1.132351508 |
| Digib | 6.199281867 | 6 | 6 | 2.256834687 |
| DigiBlen | 4.593357271 | 4 | 4 | 1.306170562 |
| Animals | 18.58737283 | 18 | 19 | 6.269373387 |
| Veg | 12.88569719 | 13 | 12 | 4.960047458 |
| Traila | 40.54697786 | 33 | 30 | 25.57298611 |
| Trailarr | 0.153650509 | 0 | 0 | 0.561471563 |
| Trailali | 23.87881508 | 24 | 24 | 1.202049764 |
| Trailb | 113.0486236 | 87 | 300 | 70.2038606 |
| Trailbrr | 0.975613405 | 0.5 | 0 | 2.167214157 |
| TrailBli | 23.26675643 | 24 | 24 | 2.978427029 |
| WAIS | 49.63749252 | 52 | 56 | 15.29008428 |
| Memunits | 11.03291442 | 12 | 0 | 5.826871113 |
| Memtime | 15.56687612 | 15 | 15 | 2.771669834 |
| Boston | 26.47112507 | 28 | 29 | 4.163700076 |
| IntraCranialVol | 1517451.388 | 1507929.049 | 1742071.545 | 183611.7777 |
| lhCortexVol | 200197.4593 | 198379.2069 | 201312.873 | 22125.86646 |
| RhCortexVol | 201808.1667 | 199844.7681 | 202468.9269 | 22045.56617 |
| CortexVol | 201808.1667 | 199844.7681 | 202468.9269 | 22045.56617 |
| SubCortGrayVol | 59903.21691 | 50681 | 54157 | 32393.84118 |
| TotalGrayVol | 543784.6603 | 539253.7999 | 539253.7999 | 54699.24728 |
| SupraTentorialVol | 913624.1604 | 906140.917 | 937093.5954 | 102152.8498 |
| LhCorticalWhiteMatterVol | 209097.0537 | 207586.004 | 195554.8444 | 30037.42343 |
| RhCorticalWhiteMatterVol | 210485.3262 | 209180.1004 | 195338.9511 | 30144.01554 |
| CorticalWhiteMatterVol | 419582.3799 | 417355.1436 | 390893.7955 | 60011.12634 |
| Mmse | 27.72770796 | 29 | 30 | 3.318152566 |
| Age At Entry | 70.965498 | 70.21218 | 66.83641 | 6.809005959 |
| Commun | 0.176615799 | 0 | 0 | 0.415457687 |
| homehobb | 0.203171753 | 0 | 0 | 0.461881072 |
| Judgment | 0.234290844 | 0 | 0 | 0.447073115 |
| Memory | 0.256582885 | 0 | 0 | 0.463565073 |
| Orient | 0.192998205 | 0 | 0 | 0.435056211 |
| Perscare | 0.092010772 | 0 | 0 | 0.34507961 |
| Apoe | 32.26870138 | 33 | 33 | 4.57917215 |
| Sumbox | 1.151331538 | 0 | 0 | 2.342166218 |
| Height | 65.75523639 | 66 | 64 | 3.72698478 |
| Weight | 168.1741472 | 165 | 160 | 36.2197191 |

## Feature selection

Though there are 39 features in the multimodal dataset, all these features will not provide same importance to the prediction. To identify the important features, feature selection operation is performed on both individual dataset and multimodal dataset. For feature selection, the Pearson's Correlation is used.

Pearson's correlation presented in Fig 1 is used to determine whether or not two quantitative variables have a linear connection. As it is based on the method of covariance, it is known as the best approach of quantifying the relationship between variables. It indicates the size of the correlation as well as the direction of a linear relationship. Besides, cross-validated Recursive Feature Elimination (RFE) and Boruta feature selection technique is also used initially for selecting important features. But it is found that these two approaches has its own drawbacks. RFE is computationally expensive and after running the RFE each time the number of selected features for ML models keep changing. So, for this research RFE was not a suitable feature selection technique. Besides, Boruta was not providing high accuracy score. Depending on these drawbacks, Pearson's correlation is preferred in this research work.

Table 5 shows various features of three individual datasets which are not important (will be removed) based on different threshold levels. Here, threshold values are resembling the degree of similarity between two features. This values range between -1.0 and 1.0. 1.0 means completely correlated -1.0 means no correlation. In feature selection, there is no need to work on two features which are strongly correlated. One feature between these two should be selected to reduce the computation complexity and to minimize the errors. The best performing threshold value is 0.9 for Clinical dataset. For MRI dataset, the best value is received on 0.9 and 0.95 threshold. For Psychological dataset, the best performance is received on 0.75, 0.8, 0.85, and 0.9 threshold. Non influential features are also selected for multimodal dataset which are mentioned in the Table 6. After removing CortexVol, CorticalWhiteMatterVol, TotalGrayVol, RhCortexVol, and RhCorticalWhiteMatterVol the best performing model is found. for these features, the threshold value was 0.95.

## Model implementation

It is already mentioned that this medical dataset doesn't contain the same number of data for each five classes. For creating a balanced dataset, Synthetic Minority Oversampling Technique (SMOTE) oversampling is used here. One of the most popular oversampling techniques to address the imbalance issue is SMOTE. By creating minority class samples at random and duplicating them, it seeks to adjust the distribution of classes. SMOTE creates new minority instances by combining minority instances that already exist. After oversampling each five classes contains 2248 instances.

As RF is the best-performing model, the implementation details of RF is mentioned here. The decision tree classifier serves as the base estimator in this case. Every estimator learns from a unique bootstrap sample taken from the training set. All features are used by estimators for training and prediction. There are 100 trees in the forest. Gini is a metric used to assess the quality of a split. A minimum of two samples are required to split an internal node. Nodes are enlarged until all leaves have fewer than two samples. A leaf node requires a minimum of one sample to be present. When determining the optimal split, the square root function is employed to calculate the number of attributes to examine. The maximum depth of the tree is 20 and the minimum one node can be present as a leaf. The Best-first algorithm is used to grow the tree. Best nodes are characterized by their relative impurity reduction. When constructing trees, bootstrap samples are used. Because the random state is preserved at 42, the model receives the same test and train sets on each execution. Weight one is meant to be assigned to all classes.

The training, testing, and 10-fold cross-validation accuracy are presented in the result section. 80% data is kept as a training set and the rest is kept as a testing set. For the stratified k-fold cross-validation procedure, the input dataset is split into k groups of samples with equal sizes. These samples are known as folds. The prediction function uses k-1 folds for each

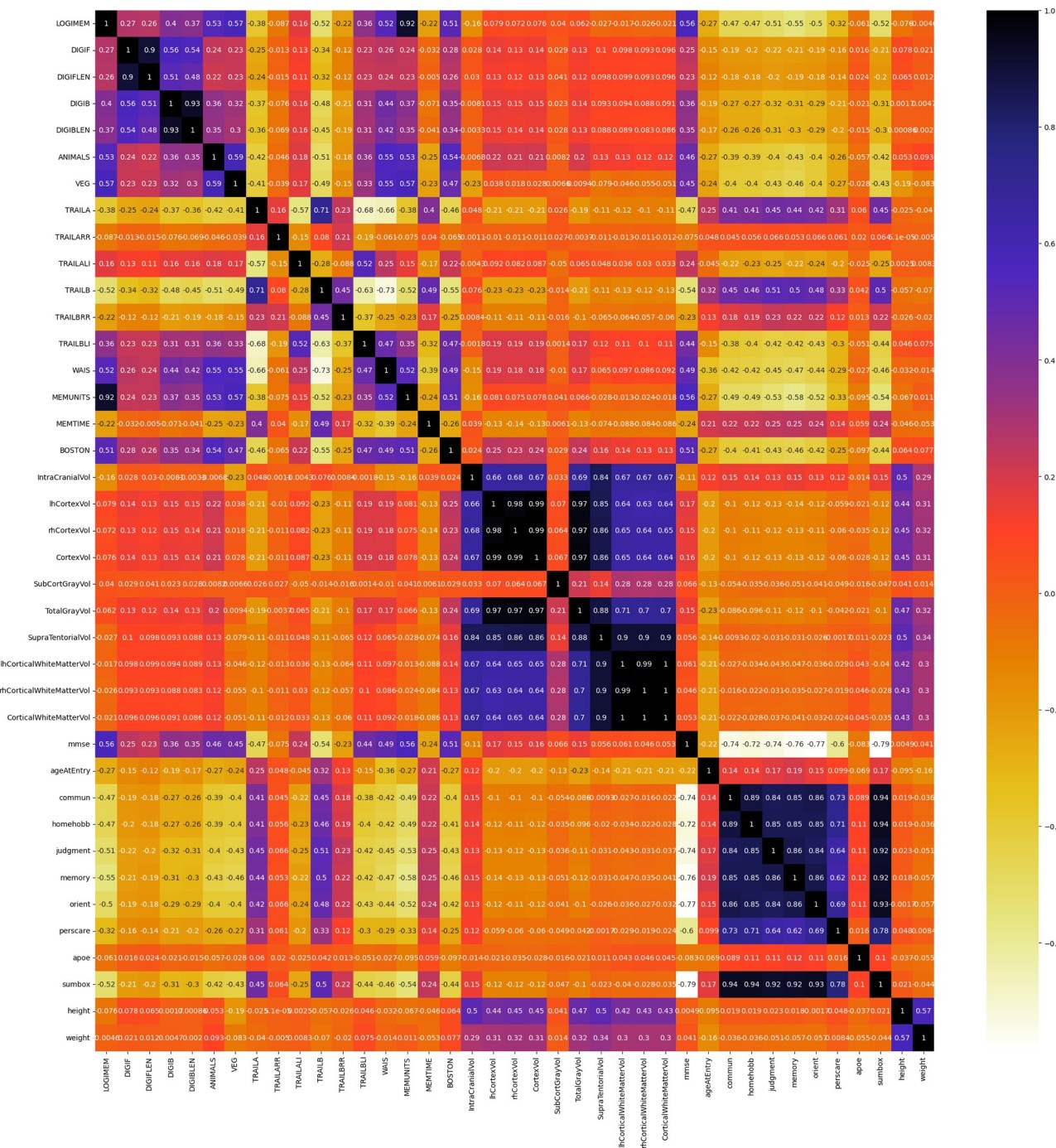

**Fig 1. Pearson's correlation heatmap of 39 features.**

learning or training set, while the remaining folds are used as a testing set. To make sure that each fold of the dataset contains the same number of observations with a certain label, stratified K-fold cross-validation is performed. Stratified Cross-validation with 10 folds is employed in this study. The dataset is then divided into ten folds. The first fold is saved for testing the model on the first iteration, while the remaining folds are used to train the model. The second

**Table 5. Features selection on individual dataset.**

| Dataset | Threshold | Features will be removed |
|---|---|---|
| Clinical | 0.7 | Commun, Homehobb, Judgment, Memory, Orient, Perscare, and Sumbox |
| | 0.75 | Homehobb, Judgment, Memory, Orient, and Sumbox |
| | 0.8 | Homehobb, Judgment, Memory, Orient, and Sumbox |
| | 0.85 | Homehobb, Memory, Orient, and Sumbox |
| | **0.9** | Sumbox |
| | 0.95 | - |
| MRI Segmentation | 0.7 | CortexVol, CorticalWhiteMatterVol, SupraTentorialVol, TotalGrayVol, LhCorticalWhiteMatterVol, RhCortexVol, and RhCorticalWhiteMatterVol |
| | 0.75 | CortexVol, CorticalWhiteMatterVol, SupraTentorialVol, TotalGrayVol, LhCorticalWhiteMatterVol, RhCortexVol, and RhCorticalWhiteMatterVol |
| | 0.8 | CortexVol, CorticalWhiteMatterVol, SupraTentorialVol, TotalGrayVol, LhCorticalWhiteMatterVol, RhCortexVol, and RhCorticalWhiteMatterVol |
| | 0.85 | CortexVol, CorticalWhiteMatterVol, SupraTentorialVol, TotalGrayVol, LhCorticalWhiteMatterVol, RhCortexVol, and RhCorticalWhiteMatterVol |
| | **0.9** | CortexVol, CorticalWhiteMatterVol, TotalGrayVol, RhCortexVol, and RhCorticalWhiteMatterVol |
| | **0.95** | CortexVol, CorticalWhiteMatterVol, TotalGrayVol, RhCortexVol, and RhCorticalWhiteMatterVol |
| Psychological | 0.7 | DigiBlen, DigiFlen, Memunits, TrailB, and WAIS |
| | **0.75** | DigiBlen, DigiFlen, and Memunits |
| | **0.8** | DigiBlen, DigiFlen, and Memunits |
| | **0.85** | DigiBlen, DigiFlen, and Memunits |
| | **0.9** | DigiBlen, DigiFlen, and Memunits |
| | 0.95 | - |

fold is used to test the model on the second iteration, while the remaining folds are utilized to train the model. Until no fold is left or reserved for the test fold, this process will be repeated. In each iteration, the testing result will be saved. After the 10 iterations, all the results are averaged to find the model's final result.

**Table 6. Features selection on multimodal dataset.**

| Threshold | Features will be removed |
|---|---|
| 0.7 | CortexVol, CorticalWhiteMatterVol, DigiBlen, DigiFlen, Memunits, SupraTentorialVol, TRAILB, TotalGrayVol, WAIS, Commun, Homehobb, Judgment, LhCorticalWhiteMatterVol, Memory, Orient, Perscare, RhCortexVol, RhCorticalWhiteMatterVol, and Sumbox |
| 0.75 | CortexVol, CorticalWhiteMatterVol, DigiBlen, DigiFlen, Memunits, SupraTentorialVol, TotalGrayVol, Homehobb, Judgment, LhCorticalWhiteMatterVol, Memory, Orient, RhCortexVol, RhCorticalWhiteMatterVol, and Sumbox |
| 0.8 | CortexVol, CorticalWhiteMatterVol, DigiBlen, DigiFlen, Memunits, SupraTentorialVol, TotalGrayVol, Homehobb, Judgment, LhCorticalWhiteMatterVol, Memory, Orient, RhCortexVol, RhCorticalWhiteMatterVol, and Sumbox |
| 0.85 | CortexVol, CorticalWhiteMatterVol, DigiBlen, DigiFlen, Memunits, SupraTentorialVol, TotalGrayVol, Homehobb, LhCorticalWhiteMatterVol, Memory, Orient, RhCortexVol, RhCorticalWhiteMatterVol, and Sumbox |
| 0.9 | CortexVol, CorticalWhiteMatterVol, DigiBlen, DigiFlen, Memunits, TotalGrayVol, RhCortexVol, RhCorticalWhiteMatterVol, and Sumbox |
| **0.95** | CortexVol, CorticalWhiteMatterVol, TotalGrayVol, RhCortexVol, and RhCorticalWhiteMatterVol |

## Explainable AI model

Explainable AI is a collection of tools and frameworks designed to assist one to understand and interpret the predictions made by the ML models. It allows one to debug and improve model performance and assist to understand the behaviors of the model. SHAP is a game-theoretic technique that may be used to explain the output of any machine learning model. It ties optimal credit allocation to local explanations by employing game theory's traditional Shapley values and their related extensions.

A feature value's Shapley value is its influence to the payout, weighted and totaled across all conceivable feature value combinations:

$$\phi_j(val) = \sum_{S \subseteq \{1,\dots,p\} \setminus \{j\}} \frac{|S|!(p - |S| - 1)!}{p!} (val(S \cup \{j\}) - val(S)) \tag{1}$$

where p is the quantity of features, x is the vector of feature values for the instance that needs to be explained, and S is a subset of the features used in the model. The prediction for feature values in set S that are prioritized above features not in set S is known as val (S):

$$val_x(S) = \int \hat{f}(x_1, \dots, x_p) d\mathbb{P}_{x \notin S} - E_X(\hat{f}(X)) \tag{2}$$

One must execute numerous integrations for each missing feature S. Here's an example: testing the prediction for the coalition S comprising of feature values x1 and x3 using the machine learning model with four features x1, x2, x3, and x4:

$$val_x(S) = val_x(\{1, 3\}) = \int_{\mathbb{R}} \int_{\mathbb{R}} \hat{f}(x_1, X_2, x_3, X_4) d\mathbb{P}_{X_2 X_4} - E_X(\hat{f}(X)) \tag{3}$$

This is analogous to the linear model's feature contributions. The Shapley value is the sole attribution technique that meets the qualities Efficiency, Symmetry, Dummy, and Additivity, all of which can be used to define a fair compensation. So, the Shapley value of a variable (or multiple variables) for a specific individual is its contribution to the difference between the value predicted by the model and the mean of all individual predictions. To accomplish this:

Step 1: Determining Shapley values for a certain individual. Simulate various value combinations for the input variables.

Step 2: Measure the difference between the expected value and the average of the predictions for each combination. As a result, the Shapley value of a variable corresponds to the mean of the value contributions based on the various combinations. Tree SHAP is a fast and accurate approach for estimating SHAP values for tree models and ensembles of trees under a variety of feature dependence assumptions. All of the decision-making features and their percentages are examined and viewed in this research work using the SHAP Tree Explainer. Through this technique, a doctor or a patient can easily understand the reason for the model's decision, as well as the decision-making features and percentage.

## Proposed AD prediction model architecture

As previously stated, we have performed data-level fusion on three different data domains which are clinical data, MRI segmentation data, and psychological assessment data. So, after completing the data-level fusion, the most popular shallow machine learning models RF, LR, Decision Tree (DT), Multi Layer Perceptron (MLP), KNN, Gradient Boosting (GB), Adaptive Boosting (AdaB), SVM, and Naive Bayes (NB) are trained and tested. It is discovered that RF

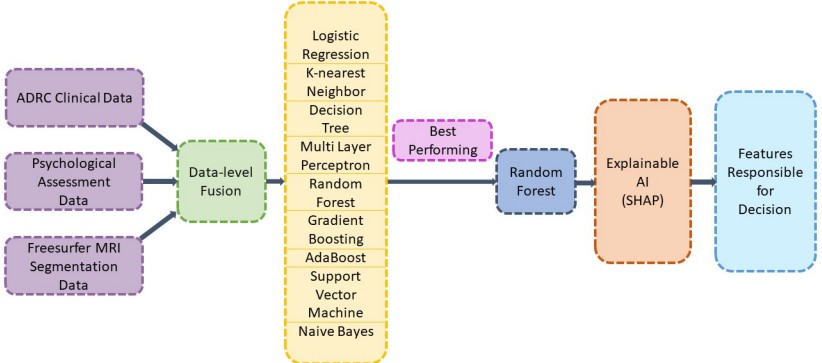

**Fig 2. Architecture of the proposed AD prediction model.**

is the best performing algorithm for five class prediction. Fig 2 shows the architecture of the classification model.

## Proposed Alzheimer's patient management framework

To provide constant observation and care to the AD positive patients, this paper proposes a framework of a GPS-based wearable sensor band that can perform management and complete monitoring. To eliminate data processing delays and provide instant responses, mist, fog, and cloud layer based patient monitoring architecture is proposed here. The system itself will choose the mist, fog, or cloud layer depending on the amount of required data processing resources. This proposed patient management architecture is depicted in detail in the Fig 3. This entire management process can be broken down into five layers, as shown below.

**Perception layer.**   The perception layer is the lower most layer, which is capable of capturing the AD patient's raw data using wearable sensor band. To make this band, one will need an optical heart rate sensor, non-invasive blood glucose monitoring sensor, gyroscope, blood pressure monitoring sensor, temperature sensor, flex sensor, and GPS tracker [21].

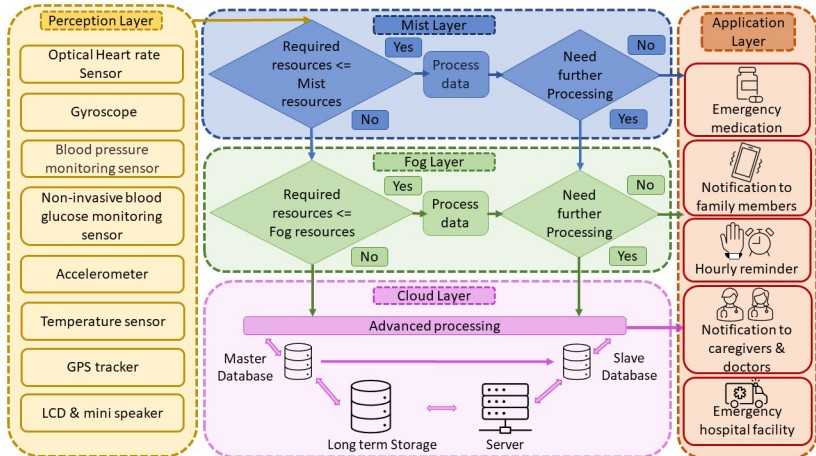

**Fig 3. Alzheimer's patient management framework.**

a. Optical heart rate sensor: An optical heart rate sensor measures pulse waves, which are variations in the volume of a blood vessel caused by the heart pumping blood. A green LED and a set of optical sensors can detect pulse waves by measuring blood volume changes. This sensor monitors the heart condition of Alzheimer's patients [22].

b. Non-invasive blood glucose monitoring sensor: The non-invasive blood glucose monitoring sensor uses photo thermal detection technique. A quantum cascade laser will emit laser and the glucose cell in blood will absorb the laser. Due to this absorption, the blood temperature will increase and this increment of temperature will detect the glucose level in blood.

c. Gyroscope: A gyroscope sensor measures and records the Orientation as well as the angular velocity of a person [23]. The MPU-6050 devices combine a 3-axis gyroscope and a 3-axis accelerometer on the same silicon die, together with an onboard Digital Motion Processor. The function of a gyroscope sensor is influenced by the core principle of momentum conservation. It works by preserving angular momentum. In a gyroscope sensor, a spinning wheel or rotor wheel is mounted on a pivot [24]. The pivot permits the rotor to rotate on a single axis, which is referred to as a gimbal.

d. Blood pressure monitoring sensor: Blood pressure sensors are pressed against the skin to calculate the pressure pulse wave and evaluate blood pressure [25].

e. Temperature sensor: Temperature sensors are used in medicine to evaluate patients' body temperatures. The working principle of temperature sensor is the voltage across the terminals of the diode [26]. The temperature rises as the voltage rises, resulting in a voltage drop between both the base transistor terminals and emitter of a diode [27].

f. Accelerometer: An accelerometer sensor is a tool that measures the acceleration of any body or object in its instantaneous rest frame. [28].

g. GPS tracker: A Global Positioning System (GPS) tracker is essential for AD patients with dementia [29]. If an Alzheimer's patient goes missing, this GPS tracker will assist to locate him [30].

h. Mini speaker: A mini speaker will assist in providing hourly reminders to the Alzheimer's patient. This mini speaker will work in tandem with the LCD to benefit Alzheimer's patients.

i. LCD: The wearable band should include a Liquid Crystal Display (LCD) to display hourly reminders. For example, medication reminders and daily activity reminders [31].

**Mist layer.** The framework now includes a mist layer for processing time-critical data of patients. Mist computing works on the network's edge with the help of actuator and sensor controllers and exists actively within the network fabric. This layer is in charge of basic sensor pre-processing such as filtering, fusion, and data aggregation. This layer will also include a comparator and decision-making process. The necessary requirements for processing sensor data are estimated after they have been pre-processed. A comparator will determine whether the required resources are equal or more than to the mist resources. If it is equal or less than, data are processed further in this layer; otherwise, data are passed to the fog layer [32]. The output of the mist layer is routed to a rule-based system, which determines the necessary activity, such as informing family members and announcing emergency medication.

**Fog layer.** The fog layer proceeds application services and computing resources closer to the edge, resulting in lower response latency. The fog layer works on those sensor data that the mist layer does not process. It also receives processed data from the mist layer for further calculation. The output of the fog layer is also routed to a rule-based system, which determines the required activity, such as hourly reminders.

**Cloud layer.** The mist, fog, and the application layer can interact with the cloud layer. Physical data from AD patients are processed in the mist and fog layers before being sent to the cloud layer for long-term storage and advanced analytics. This cloud layer is linked to a rule-based system as well.

**Application layer.** The final layer is the application layer, which displays critical information to the patient, family members, caregivers, doctors, and hospital administration. The following are some application layer features.

a. Emergency medication announcement: In critical events, the patient will receive a medication announcement. Assume the mist layer works on heart rate data from the sensor and determines the patient is in an emergency situation. The mist layer will then immediately pass these data to fog layer for more verification and the fog layer announces the required information about medicine to the patient.

b. Hourly reminder to patients: This model will provide Alzheimer's patients with hourly reminders (e.g., reminders for taking medicines, reminders for daily activity). The hourly reminder is sent to the patients' wearable band via cloud Memory.

c. Notification to doctors and caregivers: If a patient's condition worsens, caregivers and doctors will be notified. Assume that the critical patient's body sensor data requires some additional processing. In that case, the data is sent to the fog layer, and an alert is sent to the doctor and caregiver in the event of an emergency.

d. Notification to family members: Patients' family members will receive health updates. If the patient is in a critical condition, the mist layer will notify the patient's family member.

e. Emergency hospital facility: As previously stated, processed mist and fog data will be sent to the cloud layer for advance processing and long-term storage. Assume the patient is in a critical condition. In that case, the cloud will use GPS to compute the nearest hospital information and offer emergency ambulance service as well as emergency care in the hospital. Doctors and caregivers can access each patient's long-term medical records from the cloud database.

## Performance analysis

### Performance analysis using the individual dataset

Table 7 depicts the individual dataset performance for various models. For Clinical, MRI segmentation, and Psychological datasets 0.9 threshold have provided the highest accuracy. It is observed that the RF model provides best accuracy than others for all three datasets. All these results are obtained for five class classification.

### Performance analysis using the multimodal dataset

From the Table 8 it is clear that for five class prediction, RF is the best performing model if we use the multimodal dataset when the feature selection threshold is 0.95. This means if the correlation value of feature is over 0.95 then that feature is not considered. Here, the used features

**Table 7. 10 fold cross-validation accuracy of various ML models using individual datasets.**

| Dataset | Features | RF | DT | LR | KNN | MLP | GB | AdaB | SVM | XGB | NB |
|---|---|---|---|---|---|---|---|---|---|---|---|
| Clinical | Mmse, AgeatEntry, Commun, Homehobb, Judgment, Memory, Orient, Perscare, Apoe, Height, and Weight | **98%** | 95.02% | 48.6% | 88.8% | 70.15% | 48.39% | 48.39% | 45.05% | 89.02% | 62.4% |
| MRI | IntraCranial- Vol, lhCortexVol, SubCortGrayVol, SupraTentorialVol, and LhCorticalWhiteMatterVol | **88.85%** | 85.38% | 32.78% | 82.03% | 21.62% | 71.89% | 44.21% | 24.47% | 67.22% | 26.64% |
| Psychological | Logimem, DigiF, DigiB, Animals, Veg, TrailA, TrailArr, TrailAli, TrailB, TrailBrr, TrailBli, WAIS, Memtime, and Boston | **94.21%** | 81.4% | 43.01% | 85.49% | 54.6% | 82.15% | 55.9% | 26.65% | 77.17% | 36.13% |

are: Logimem, Digif, DigiFlen, Digib, DigiBlen, Animals, Veg, Traila, Trailarr, Trailali, Trailb, Trailbrr, TrailBli, WAIS, Memunits, Memtime, Boston, IntraCranialVol, lhCortexVol, SubCortGrayVol, SupraTentorialVol, LhCorticalWhiteMatterVol, Mmse, Age At Entry, Commun, Homehobb, Judgment, Memory, Orient, Perscare, Apoe, Height, Weight, and Sumbox. The training, testing, and stratified 10 fold cross validation accuracy is mentioned in Table 8. The 10 fold cross validation accuracy score of multimodal data is higher than the individual dataset. Here, the cross validation accuracy is used because it helps to avoid over-fitting of a model. From the confusion matrix shown in Fig 4 of the RF classifier we can calculate the precision, recall, and F1-score of this model. The training, testing, cross validation accuracy, precision, recall, F1-score, and area under coverage (AUC) values of the RF model are 100%, 98.84%, 98.81%, 98.94%, 98.79%, 98.75%, and 99.97%, respectively.

## Performance comparison with recent works

Table 9 shows the accuracy comparison between various recent similar works. It is observed that RF is used by El-Sappagh et al. [13]. But their dataset is different than ours. However, for doing a model to model comparison, this work is compared with ours. It is observed that the model of El-Sappagh et al. had achieved 93.95% and 87.08% accuracy in first and second layer, respectively. Whereas, our method achieved 98.81% accuracy. Using Modified RF model, Ali et al. [17] achieved 96.43% accuracy on OASIS-2 dataset. Amrutesh et al. [19] used the text data from OASIS dataset and got 92.13% accuracy on RF. Training and testing the RF model on MRI OASIS data Baglat et al. [16] and Kavitha et al. [18] achieved 86.8% and 86.92% accuracy. The version of OASIS dataset was not mentioned in their work. The accuracy of Buvari et al. [20] was 74.891% while using the MRI and segmentation dataset from OASIS-3. So, it is

**Table 8. Performance analysis on multimodal dataset.**

| Features | Performance Metrics | RF | DT | LR | KNN | MLP | GB | AdaB | SVM | NB |
|---|---|---|---|---|---|---|---|---|---|---|
| Sumbox, Memory, Judgment, Orient, Memunits, | Training Accuracy | **100%** | 100% | 30.52% | 89.13% | 25.82% | 97.69% | 54.61% | 29.47% | 40.28% |
| Logimem, SubCortGrayVol, Homehobb, Commun, | Testing Accuracy | **98.84%** | 94.217% | 31.58% | 82.206% | 25.35% | 96.307% | 54.4% | 29.53% | 40.48% |
| MMSE, TrailBrr, DigiF, TrailAli, IntraCranialVol, Veg, TrailB, DigiB, DigiFlen, | Stratified 10-fold Cross Validation Accuracy | **98.81%** | 94.92% | 31.28% | 83.825% | 25.79% | 95.65% | 55.57% | 25% | 40.24% |
| TrailBli, TrailArr, AgeatEntry, | Precision | **98.94%** | 94.51% | 30.56% | 82.35% | 46.63% | 96.32% | 52.2% | 24.06% | 41.7% |
| Perscare, SupraTentorialVol, Animals, | Recall | **98.79%** | 94.41% | 31.81% | 82.15% | 32% | 96.24% | 54.61% | 21.88% | 40.58% |
| LhCortical-WhiteMatterVol, LhCortexVol, Apoe, | F1-score | **98.75%** | 99.06% | 28.70% | 81.92% | 29% | 96.23% | 47.49% | 12.22% | 40.47% |
| DigiBlen, Height, Boston, Memtime, Weight, TrailA, and WAIS | AOU | **99.97%** | 96.5% | 63% | 95.8% | 53.4% | 99.7% | 79% | 58.3% | 75.41% |

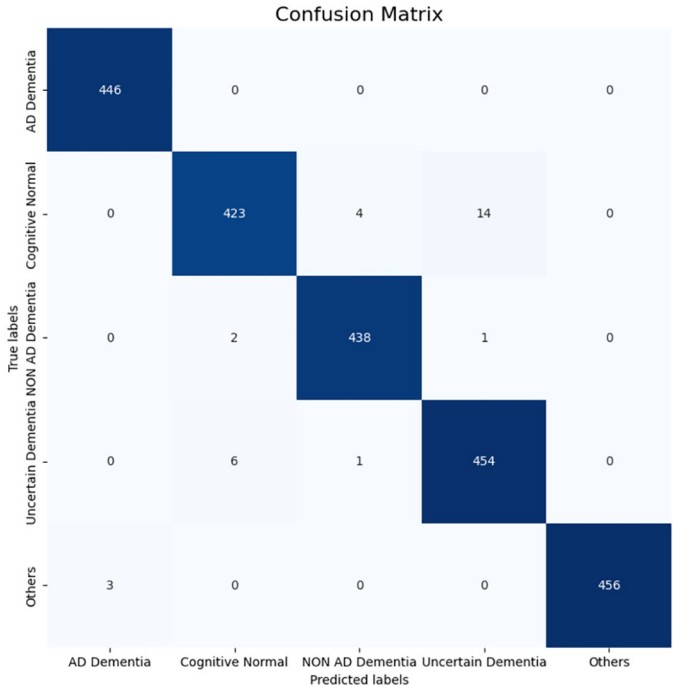

**Fig 4. Confusion matrix of the multimodal data using RF classifier.**

clear that our approach is providing better performance than others with a 10-fold cross-validation accuracy of 98.81%.

## Explain ability of the model

Fig 5 shows the five-class RF classification model's mean SHAP value of all 35 features. In this RF classification model, the top 15 influential features are Memory, Sumbox, Judgment, Homehobb, Orient, IntraCranialVol, Commun, Mmse, Perscare, Memunits, SupraTentorial-Vol, Apoe, LhCortexVol, Veg, and Logimem. As it is a multimodal dataset, among the top 15 influential features Sumbox, Memory, Judgment, Homehobb, Orient, Commun, Mmse, Perscare, and Apoe, come from the Clinical dataset. Features named Memunits, Logimem, and Veg came from the Psychological dataset. IntraCranialVol, SupraTentorialVol, and LhCortex-Vol belonged to the MRI segmentation dataset. It is also shown in Fig 5 that, for AD prediction, Homehobb is the most dominant feature. Other prominent features for AD prediction

**Table 9. Accuracy comparison between various recent similar works.**

| Author | Dataset | Model | Accuracy |
|---|---|---|---|
| El-Sappagh et al. [13] | Multi-modal data from ADNI | RF | 93.95% and 87.08% |
| Ali et al. [17] | OASIS-2 | Modified RF | 96.43% |
| Amrutesh et al. [19] | Text values from OASIS | RF | 92.13% |
| Baglat et al. [16] | MRI data from OASIS | RF | 86.8% |
| Kavitha et al. [18] | MRI data from OASIS | RF | 86.92% |
| Buvari et al. [20] | MRI and segmentation data from OASIS-3 | CNN and NN | 74.891% |
| **Our Method** | **Multi-modal data from OASIS-3** | **RF** | **98.81%** |

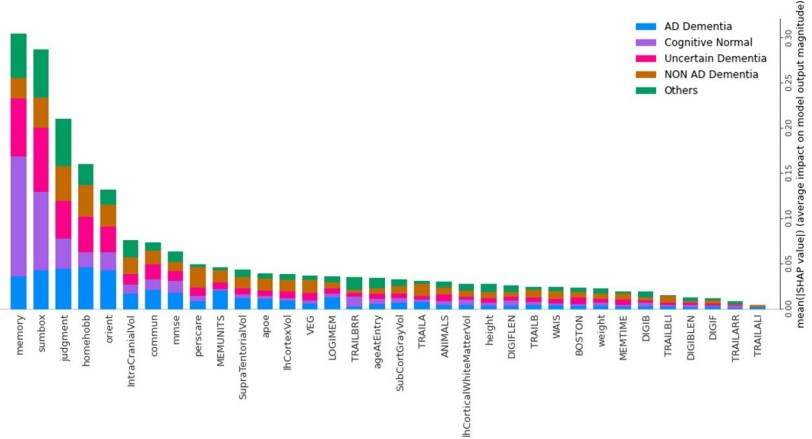

**Fig 5. Features and their influence in each class (SHAP summary bar chart).**

are Orient, Judgment, Sumbox, Memory, Memunits, Commun, MMSE, IntraCranialVol, etc. Similarly, features for other classes can also be calculated.

Table 10 presents the top 5 Features and their influence value on five classes using the RF classifier. For Memory, mean(SHAP) is about 12.33% on AD (Class 1), 42.47% on CN (Class 0), 21.92% on Uncertain (Class 3), 6.85% on Non-AD (Class 2), and 16.44% on Others (Class 4). It means Memory has more influence on predicting CN rather than the rest of the classes. For Memory, the second, third, fourth, and fifth influenced classes are Uncertain, Others, AD, and Non-AD classes, respectively. The Memory is the most influential feature among all 35 features. Similarly for Sumbox, mean(SHAP) is about 14.5% on AD (Class 1), 30.43% on CN (Class 0), 24.64% on Uncertain (Class 3), 11.6% on Non-AD (Class 2), and 18.84% on Others (Class 4). It means Sumbox has more influence on predicting CN rather than the rest of the classes. For Sumbox, the second, third, fourth, and fifth influenced classes are Uncertain, Others, AD, and Non-AD classes, respectively. Also, Sumbox is the second most influential feature among all 35 features. The importance and influence of other features can also be described in a similar manner.

**Class specific features.** Each dot in Fig 6 indicates how a specific feature for a given instance affects a specific class. Here, it is shaded based on how much each feature contributes to the overall influence of the model. Red denotes a high feature value, purple is a mid-range value, and blue is a low feature value. According to how it affects the model for that particular class, each feature is listed. A feature has a positive impact on the model for that class if its tail extends farther to the right from the neutral position (0.0). The negative impact on the model is represented by feature values from 0.0 to the left. To ensure simplicity here in Table 6 only positively impactful features are mentioned. By knowing positively impactful features, one can

**Table 10. Top 5 features and their percentage of each class for the RF classifier model.**

| Features | AD | CN | Uncertain | Non-AD | Others |
|---|---|---|---|---|---|
| Memory | 12.33% | 42.47% | 21.92% | 6.85% | 16.44% |
| Sumbox | 14.5% | 30.43% | 24.64% | 11.6% | 18.84% |
| Judgement | 21% | 16% | 20% | 18% | 24% |
| Homehobb | 28.57% | 10.39% | 24.67% | 23.37% | 12.99% |
| Orient | 29.85% | 15% | 28.9% | 16.42% | 12% |

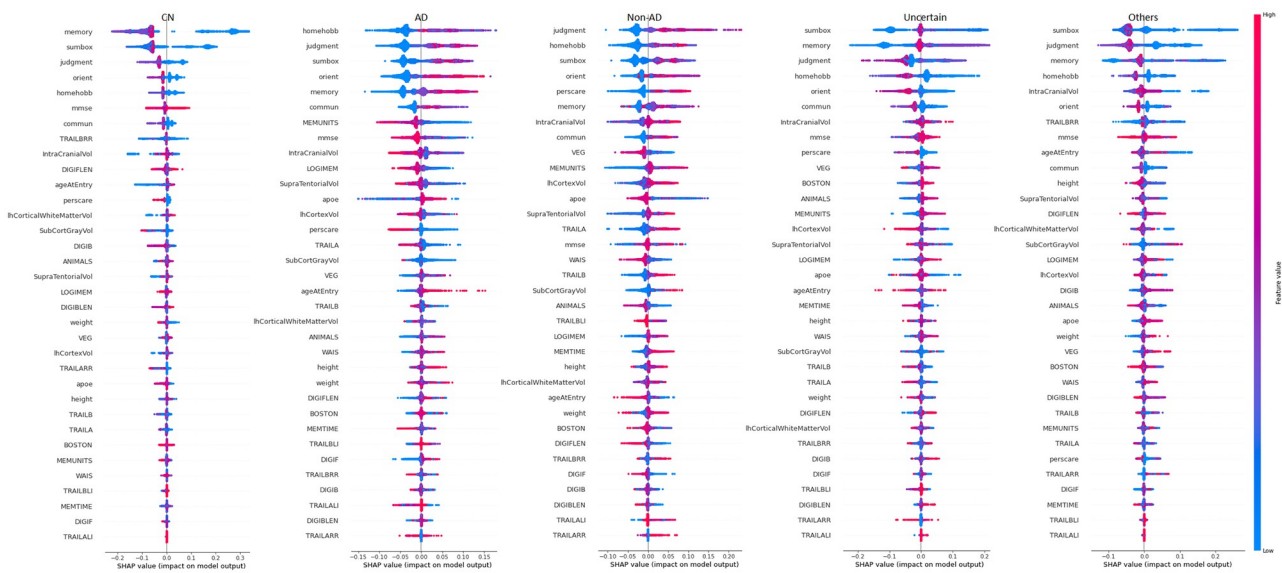

**Fig 6. Features and their influence for Five classes (SHAP dot plot).**

automatically identify negatively impactful features. Besides, Table 11 also shows the reasons behind the classification of each class. Understanding the specific function of each feature both by itself and in conjunction with other characteristics is therefore crucial for comprehending the class.

**Table 11. Name of the features which are positively influencing the RF model for each class.**

| Class | High | Low | Identify |
|---|---|---|---|
| CN | MMSE and DigiFlen | Memory, Sumbox, Judgment, Orient, Homehobb, TrailBrr, and Weight | Tail of TrailBrr in the negative side is larger than the positive side |
| AD | Homehobb, Judgement, Sumbox, Orient, Memory, AgeatEntry, Commun, Veg, Animals, TrailBli, DigiFlen, and WAIS | Memunits, MMSE, Logimem, IntraCranialVol, SupraTentorialVol, Perscare, LhCortexVol, SubCortGrayVol, TrailA, and TrailB | High value on the positive side: Veg, Animal, TrailBli, DigiF, WAIS. Low value on the positive side: Memunits, MMSE, Logimem, Boston, and TrailA. Low value of Apoe in the negative side. Besides, Homehobb and Judgment are common for AD and Non-AD. So, if the Homehobb is the best influencing feature then the class will be AD. |
| Non-AD | Judgment, Homehobb, Sumbox, Orient, Perscare, Memory, Commun, Memunit, MMSE, TrailA, TrailB, LhCortexVol, SubCortGrayVol, Memtime, TrailBrr, TrailAli, and TrailArr | Veg, Apoe, and WAIS | Judgment, Perscare, Memory, and Orient are common for AD and Non-AD. So, if the Judgment value is very much positively influenced then it will be Non-AD Class. The long tail of the lower value of Apoe in the positive side will increase the risk of Non-AD. |
| Uncertain | IntraCranialVol, DigiB, and DigiBlen | Sumbox, Homehobb, and Judgment | Features such as Sumbox and Memory have very high feature values in the neutral position (0.0). On the positive side of the axis, Memory magnitude is in mid-level (purple colored). |
| Others | MMSE, DigiB, DigiBlen, and Apoe | TrailBrr and IntraCranialVol | DigiB, and DigiBlen are common for Uncertain and Others. If the value of DigiB and DogibLen are in Mid-level (purple) then it is Others. Otherwise, they will be on a High level. IntraCranialVol is common for Others and CN. If IntraCranialVol is positively influenced then it is the Others class. |

For CN the feature values of Memory, Sumbox, Judgment, Orient, Homehobb, and Perscare are low and for these features, the model is positively influenced. Besides, MMSE and DigiFlen values are high on the positive side. Using these characteristics CN class can be classified easily. Homehobb, Judgement, Sumbox, Orient, Memory, AgeatEntry, Commun, Veg, Animals, TrailBli, DigiFlen, and WAIS are the key influenced features for prediction AD and feature value of these are High and the model is a positive influence for these features. Very highly positive feature values of Memory and Orient are the key identifying factors for AD. So, to uniquely identify AD following a combination of features are important. High value on the positive side: Veg, Animal, TrailBli, DigiF, WAIS. Low value on the positive side: Memunits, MMSE, Logimem, Boston, and TrailA. The low value of Apoe in the negative side.

In order to identify the Non-AD class, the following combination is appropriate. The high value of Judgment, Homehobb, Sumbox, Orient, Perscare, Memory, Commun, Memunit, MMSE, TrailA, TrailB, LhCortexVol, SubCortGrayVol, Memtime, TrailBrr, TrailAli, and TrailArr will influence the model in a positive way. The low values of Veg, Apoe, AgeatEntry, and WAIS are also influential for Non-AD prediction. Here, some features are common for AD and Non-AD prediction e.g., Sumbox, Memory, Homehobb, Judgement, and Orient. If the best influential feature is Judgement then the class will be Non-AD. To uniquely identify Non-AD following combination of features are important. High value on the positive side: Memunit, LhCortexVol, MMSE, TrailB. A low value on the positive side: Apoe, Veg, and WAIS. The Low value of Apoe on the positive side will increase the risk of Non-AD.

The class Uncertain can be very easily identified if the very high magnitude of Sumbox and Memory is in the neutral position (0.0). Besides the magnitude value of Memory in the positive axis is purple in color which means this magnitude is at Mid level. The class Others can be identified if the TrailBrr has a low magnitude on both positive sides. Whereas TrailBrr has a larger tail on the negative side for the CN class. Others and CN both have IntraCranialVol. If IntraCranialVol is positively modified, the Others class is affected.

**Features for individual Participants.** Waterfall plots are intended to show justifications for certain predictions on any specific participant or subject. The expected feature value of the model output appears at the bottom of a waterfall plot, and each row demonstrates how the negative (blue) or positive (red) contribution of each feature changes the value from the expected results $E[f(x)]$ over the dataset to the model output $f(x)$ for this prediction.

In Fig 7 it is interesting that having the value of Memory = 0, Sumbox = 0, Orient = 0, Judgment = 0, Homehobb = 0, Commun = 0, MMSE = 24.5, IntraCranialVol = 1727701.242, and DigiB = 5; increases this particular person's probability to be in CN class. Here, each feature is contributing positively and the overall prediction score is 99%. Fig 8 shows that the model's prediction score for classifying AD is 98%. Here, the important features and their scores are Memory = 1, Sumbox = 3.513, Judgment = 0551, Homehobb = 0.654, DigiFlen = 6.103, Orient = 0.551, Commun = 0.654, Height = 70.974, and Age at entry = 71.704. These features are contributing positively.

Similarly, for Non-AD classification in Fig 9, Judgment, Perscare, Homehobb, Sumbox, IntraCranialVol, etc features are positively influencing the model and the prediction accuracy is 96%. The Figs 10 and 11 are showing the features and their positive influence on Uncertain and Others, respectively. These models' prediction values are 97% and 97% for Uncertain and Others, respectively.

## Clinical evidence

To understand and measure level the significance of the conclusion that Memory, Sumbox, Judgment, Homehobb, and Orient are the most important features for predicting five class

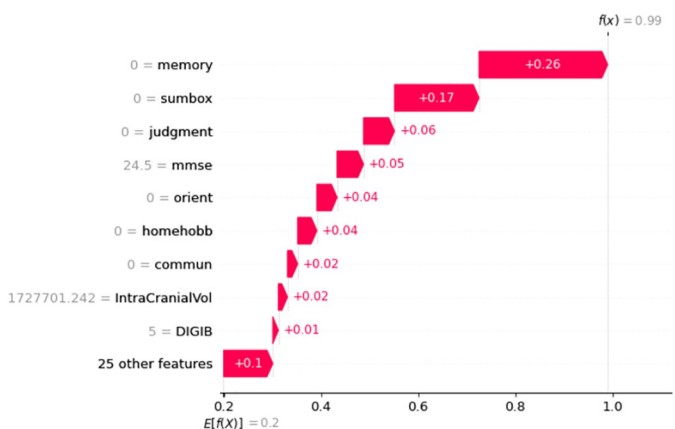

**Fig 7. Features and their influence for a participant with CN class (SHAP waterfall plot).**

predictions, the t-test is done and the p-value is calculated. Here, the Null Hypothesis was —"Memory, Sumbox, Judgment, Homehobb, and Orient are not contributing to predict AD." The Alternative Hypothesis was—"Memory, Sumbox, Judgment, Homehobb, and Orient are contributing to predict AD". For all classes, the p-values of these features were less than 0.001. For the AD class, the p-values of Memory, Sumbox, Judgment, Homehobb, and Orient are 0.079,.00000091, 0.00064, 0.00072, and 0.0000095 respectively. As the p-values are less than the significance value of 0.05, it can be stated that the Null Hypothesis is rejected and the Alternative Hypothesis is accepted. So, now it is evident that Memory, Sumbox, Judgment, Homehobb, and Orient are the most contributing factors for predicting AD.

## Conclusion

This study proposes an explainable multimodal approach for predicting AD. This multimodal approach consists of data-level fusion on three datasets from three different genres. ADRC clinical data, MRI segmentation numeric data, and psychological data from the OASIS-3 repository are used here. The feature selection operation is performed here to improve the ML

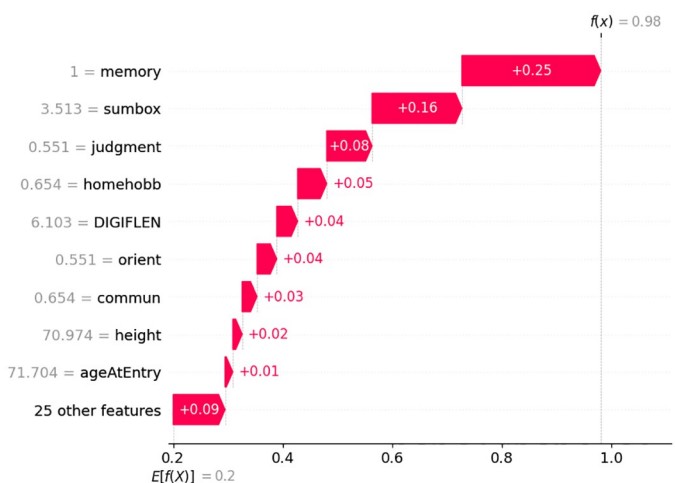

**Fig 8. Features and their influence for a participant with AD class (SHAP waterfall plot).**

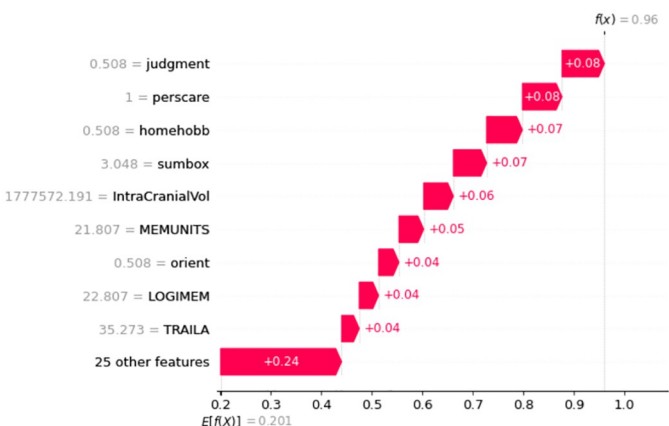

**Fig 9. Features and their influence for a participant with Non-AD class (SHAP waterfall plot).**

process and increase the predictive power of ML algorithms by finding the important variables and eliminating the redundant and irrelevant features. This multimodal approach provides 98.81% accuracy for five classifications using an RF classifier. Whereas using clinical, psychological, and MRI segmentation data the achieved accuracies are 98.0%, 94.21%, and 88.85%. So it is clear that the usage of multimodal dataset can bring a better-performing AD prediction model. To achieve the trustworthiness of the predictions of this model, the SHAP explainer is used here, and all the decision-making features along with their values are displayed. From the outcome of the explainer, it is clear that Judgment, Memory, Homehobb, Orient, and Sumbox are the most important features. Even for perfect decision-making, features of all three individual datasets played an important role. This work also provided an entire architecture for AD patient management and 24/7 monitoring. One limitation of this work is that the effectiveness of the proposed AD patient management architecture is not evaluated. In the future, this proposed patient management system should be implemented and evaluated in real life. An efficient model can be developed that will give early AD prediction and prediction

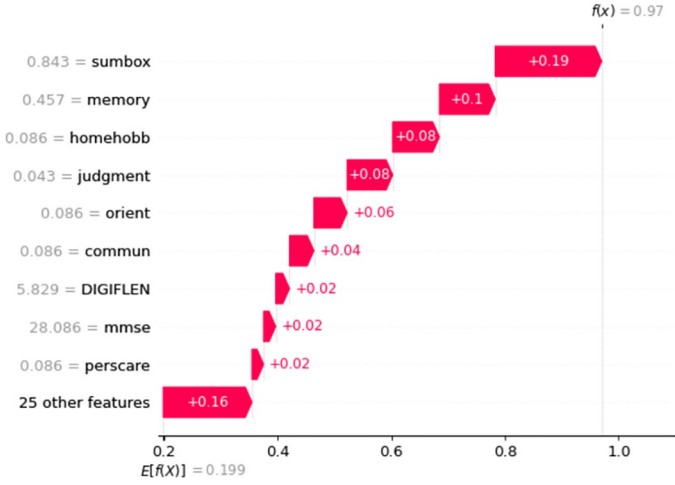

**Fig 10. Features and their influence for a participant with Uncertain class (SHAP waterfall plot).**

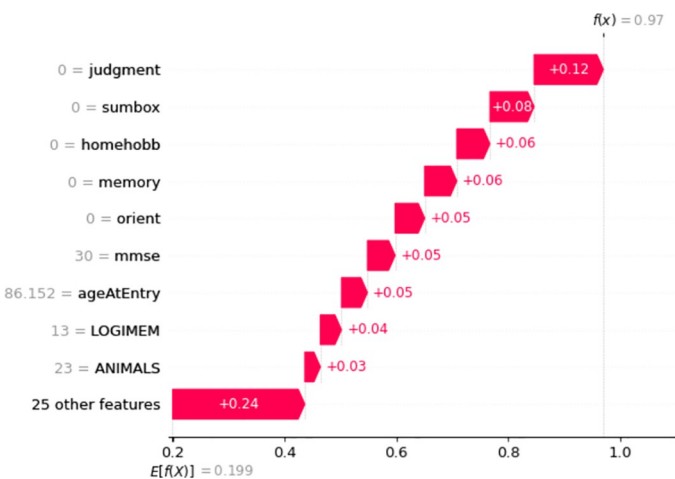

**Fig 11. Features and their influence for a participant with Others class (SHAP waterfall plot).**

on different AD stages. To ensure the multimodality of the dataset at a robust level, demographic data, text data, numerical data, and image data could also be used.

## Acknowledgments

The authors want to acknowledge sincere gratitude to the ICT Division of the Government of the People's Republic of Bangladesh, the Kunsan National University, the Woosong University, the Jahangirnagar University, and the Bangladesh University of Professionals.

## Author Contributions

**Conceptualization:** Sobhana Jahan, Mufti Mahmud, Md. Sazzadur Rahman.

**Data curation:** Sobhana Jahan.

**Formal analysis:** Sobhana Jahan, A. S. M. Sanwar Hosen.

**Funding acquisition:** In-Ho Ra.

**Investigation:** Sobhana Jahan, M. Shamim Kaiser, Md. Sazzadur Rahman.

**Methodology:** Sobhana Jahan, Kazi Abu Taher, M. Shamim Kaiser, Mufti Mahmud, Md. Sazzadur Rahman, A. S. M. Sanwar Hosen.

**Resources:** Kazi Abu Taher.

**Supervision:** M. Shamim Kaiser.

**Validation:** Sobhana Jahan, Md. Sazzadur Rahman.

**Visualization:** Sobhana Jahan.

**Writing – original draft:** Sobhana Jahan.

**Writing – review & editing:** Kazi Abu Taher, M. Shamim Kaiser, Mufti Mahmud, Md. Sazzadur Rahman, A. S. M. Sanwar Hosen, In-Ho Ra.

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
