## [Decision Letter · Decision Letter 0]

3 Aug 2023

PONE-D-23-09483Explainable AI-based Alzheimer’s Prediction and Management Using Multimodal DataPLOS ONE

Dear Dr. Kaiser,

Thank you for submitting your manuscript to PLOS ONE. After careful consideration, we feel that it has merit but does not fully meet PLOS ONE’s publication criteria as it currently stands. Therefore, we invite you to submit a revised version of the manuscript that addresses the points raised during the review process.

Please see the comments of reviewers 3 and 4 below. Please ignore the comments from reviewer 1 and 2, as these reviews do not comply with PLOS ONE policy. In addition, please note that there is no requirement from PLOS ONE to cite any of the works recommended by the reviewers. Please also note that reviewer 4's concerns about the quality of the figures can be ignored - the low quality only shows up in the PDF generation and will be fine for eventual publication.

We look forward to receiving your revised manuscript.

Kind regards,

Hanna Landenmark

Staff Editor

PLOS ONE

Journal Requirements:

"This work is supported by the research funds of the Information and Communication Technology

division of the Government of the People’s Republic of Bangladesh in 2020 - 2021, the National Research Foundation of Korea (NRF) Grant by the

Korean Government through the Ministry of Science and ICT (MSIT) under Grant 2021R1A2C2014333,

and the Woosong University Academic Research in 2022."

4. Please upload a new copy of Figure 6 as the detail is not clear. Please follow the link for more information: " ext-link-type="uri" xlink:type="simple">https://blogs.plos.org/plos/2019/06/looking-good-tips-for-creating-your-plos-figures-graphics/"
https://blogs.plos.org/plos/2019/06/looking-good-tips-for-creating-your-plos-figures-graphics/.

Reviewers' comments:

Reviewer's Responses to Questions

**Comments to the Author**

1. Is the manuscript technically sound, and do the data support the conclusions?

Reviewer #1: Partly

Reviewer #2: Partly

Reviewer #3: Yes

Reviewer #4: Yes

2. Has the statistical analysis been performed appropriately and rigorously? 

Reviewer #1: No

Reviewer #2: Yes

Reviewer #3: Yes

Reviewer #4: Yes

3. Have the authors made all data underlying the findings in their manuscript fully available?

Reviewer #1: Yes

Reviewer #2: Yes

Reviewer #3: Yes

Reviewer #4: Yes

4. Is the manuscript presented in an intelligible fashion and written in standard English?

Reviewer #1: No

Reviewer #2: No

Reviewer #3: Yes

Reviewer #4: Yes

5. Review Comments to the Author

Reviewer #1: The overall impression of the technical contribution of the current study is reasonable. However, the Authors may consider making necessary amendments to the manuscript for better comprehensibility of the study.

1. The abstract must be re-written, focusing on the technical aspects of the proposed model, the main experimental results, and the metrics used in the evaluation. Briefly discuss how the proposed model is superior.

2. Additionally, method names should not be capitalized. Moreover, it is not the best practice to employ abbreviations in the abstract, they should be used when the term is introduced for the first time.

3. Introduction section must discuss the technical gaps associated with the current problem.

4. The section Introduction is not given the index, where as the related work section is given the index number 1. Author must make sure the consistency is maintained.

5. Authors are recommended to incorporate some of the relevant studies like https://doi.org/10.3390/electronics11244086 and https://doi.org/10.1155/2022/8167821

6. Authors may provide the architecture/block diagram of the proposed model for better comprehensibility of the proposed model concerning various aspects of the proposed model.

7. More explanation of the proposed model is desired on technical grounds.

8. The important details, like the input/tensor/kernel size, must be discussed, and whether authors have used Stride 1 or Stride, 2 must be presented. What type of activation function is being used in the current study. For better idea refer and include https://doi.org/10.1007/s12652-021-03612-z

9. For how many epochs does the proposed model execute. what is the initial learning rate, and after how many epochs does the model's learning rate saturated.

10. Authors may show some mathematical equations towrds the section. "Missing Data Imputation Using KNN "

11. Authors may provide the citations for the values (Mean, Median, Mode and SD) that are shown in Table 4.

12. Majority of the figures lack the clarity, they quality is fair but they must be explained in the text and the figures must be cited. Where is the graph for testing loss and accuracy presented in the study.(Add hyperparameters if it is feasible to the authors)

13. Please discuss more on the implementation platform.

14. What are the cases assumed as TP, TN, FP, FN (confusion matrix) in the current study, For better idea refer and include https://doi.org/10.3390/cancers14174191

15. More comparative analysis with state-of-art models is desired.

16. By considering the current form of the conclusion section, it is hard to understand by PLOSOne Journal readers. It should be extended with new sentences about the necessity and contributions of the study by considering the authors' opinions about the experimental results derived from some other well-known objective evaluation values if it is possible.

Reviewer #2: 1) Abstract Lack of context and clarity: The abstract doesn't explain the study's motivation and significance work. The abstract must be re-written focusing on the technical aspects of the proposed model, and the metrics used in the evaluation. Briefly discuss how the proposed model is superior.

2) The contribution of the current study must be clearly discussed. And motivation must also be discussed in the manuscript. The authors would have to explain their technical contributions better and more clearly.

3) The conclusion does not include all details of results and comparisons, and it should be completely explained by more details.

4) Authors should mention to the Novelty of the work with respect to existing work.

5) Authors should provide Methodology section and explain all algorithms by more correct details not briefly in that section. For example, RF, DT, KNN, and other used algorithms in this manuscript are not explained.

6) Introduction section must discuss the technical gaps associated with the current problem.

7) The details inside figures do not have good resolution, and they are not readable. The author should modify them to legible figures for readers.

Reviewer #3: The content is concise, clear-structured, detailed, and it’s certainly appropriate for being published in PLOS ONE. My recommendation is a minor revision. Below are some suggestions to authors:

Could the author elaborate the reason why deep learning models are excluded from the analysis? Neural networks can achieve both feature selection and outcome classification at the same time. Actually, other than the two studies mentioned in the manuscript (Buvari et al. and Kumari et al.), other studies also demonstrated the strength of neural networks on capturing data structure and predicting AD with high accuracy. NNs are one of the famous methods applied to AD data. Considering that the authors using a multimodal dataset, I would recommend looking into the hybrid NN, i.e., the multimodal neural network, which joins information from different modalities to perform a prediction task. Additionally, it seems like the accuracy of RF is too good to be true. I wonder are all the results displayed in the manuscript from the testing data not training data? And did the authors check for over-fitting issue? It would be nice if the authors add the details of the training process.

One last small thing is some abbreviations’ full names appear multiple times in the manuscript, such as ADRC, or appear before the full name, such as OASIS.

Thanks,

Reviewer #4: This paper proposes a novel explainable AI-based Alzheimer's disease (AD) prediction model using a multimodal dataset. The authors perform data-level fusion using clinical data, MRI segmentation data, and psychological data to predict five-class classification of AD. They evaluate the performance of nine popular machine learning models and find that Random Forest achieves the highest accuracy. The authors also utilize the SHAP model for explainability and analyze the causes of prediction.

1.Limited evaluation: The paper lacks a thorough evaluation and comparison of the proposed method with existing state-of-the-art models or baselines. Have you compared the performance of your proposed method with other state-of-the-art models or baselines? If not, why not? The authors have provided a list of only 9 widely-used machine learning models; however, it should be noted that not all of these models perform at a satisfactory level.

2.Lack of implementation details: The paper does not provide sufficient details on the implementation of the proposed method, making it difficult for readers to reproduce the study. Can you provide more details on the implementation of the proposed method, such as the specific parameters used for each machine learning model?

3.Alzheimer’s Prediction using AI is a hot topic. To enhance interest to the readers, it is better to give a more extensive review on this topic. There are some representative works, such as:

Morphological Feature Visualization of Alzheimer's Disease via Multidirectional Perception GAN;

Predicting clinical scores for Alzheimer's disease based on joint and deep learning;

Tensorizing GAN With High-Order Pooling for Alzheimer's Disease Assessment.

4.Given that the paper focuses on explainability and analysis of prediction causes, it would enhance the overall contribution if there were clinical evidence to support the conclusions.

5.The figures presented in the paper appear to be blurry and do not meet the standards for effective visual presentation. To improve resolution and visual quality, it is recommended that the authors utilize vector graphics.

6. PLOS authors have the option to publish the peer review history of their article (what does this mean?). If published, this will include your full peer review and any attached files.

Reviewer #1: No

Reviewer #2: **Yes: **Zari

Reviewer #3: **Yes: **Yizhuo Wang

Reviewer #4: No

---

## [Decision Letter · Decision Letter 1]

30 Oct 2023

Explainable AI-based Alzheimer’s Prediction and Management Using Multimodal Data

PONE-D-23-09483R1

Dear Dr. ,

We’re pleased to inform you that your manuscript has been judged scientifically suitable for publication and will be formally accepted for publication once it meets all outstanding technical requirements.

Kind regards,

M. Firoz Mridha

Academic Editor

PLOS ONE

Additional Editor Comments (optional):

Reviewers' comments:

Reviewer's Responses to Questions

**Comments to the Author**

1. If the authors have adequately addressed your comments raised in a previous round of review and you feel that this manuscript is now acceptable for publication, you may indicate that here to bypass the “Comments to the Author” section, enter your conflict of interest statement in the “Confidential to Editor” section, and submit your "Accept" recommendation.

Reviewer #3: All comments have been addressed

Reviewer #4: All comments have been addressed

2. Is the manuscript technically sound, and do the data support the conclusions?

Reviewer #3: Yes

Reviewer #4: Yes

3. Has the statistical analysis been performed appropriately and rigorously? 

Reviewer #3: Yes

Reviewer #4: Yes

4. Have the authors made all data underlying the findings in their manuscript fully available?

Reviewer #3: Yes

Reviewer #4: Yes

5. Is the manuscript presented in an intelligible fashion and written in standard English?

Reviewer #3: Yes

Reviewer #4: Yes

6. Review Comments to the Author

Reviewer #3: (No Response)

Reviewer #4: My previous concerns have been well addressed. I recommend its acceptation. Besides, some figures are unclear, it is better to improve.

7. PLOS authors have the option to publish the peer review history of their article (what does this mean?). If published, this will include your full peer review and any attached files.

Reviewer #3: **Yes: **Yizhuo Wang

Reviewer #4: No

---

## [Editor Report · Acceptance letter]

7 Nov 2023

PONE-D-23-09483R1 

Explainable AI-based Alzheimer’s Prediction and Management Using Multimodal Data 

Dear Dr. Kaiser:

I'm pleased to inform you that your manuscript has been deemed suitable for publication in PLOS ONE. Congratulations! Your manuscript is now with our production department. 

Kind regards, 

on behalf of

Dr. M. Firoz Mridha 

Academic Editor

PLOS ONE